# Contribution of *Lactobacilli* on Intestinal Mucosal Barrier and Diseases: Perspectives and Challenges of *Lactobacillus casei*

**DOI:** 10.3390/life12111910

**Published:** 2022-11-16

**Authors:** Da Qin, Yixuan Ma, Yanhong Wang, Xilin Hou, Liyun Yu

**Affiliations:** 1Heilongjiang Provincial Key Laboratory of Environmental Microbiology and Recycling of Argo-Waste in Cold Region, College of Life Science and Biotechnology, Heilongjiang Bayi Agricultural University, Daqing 163319, China; 2Colleges of Animal Science and Technology, Heilongjiang Bayi Agricultural University, Daqing 163319, China

**Keywords:** *Lactobacillus*, intestinal mucosal barrier, GIT-related diseases, innate immunity

## Abstract

The intestine barrier, the front line of normal body defense, relies on its structural integrity, microbial composition and barrier immunity. The intestinal mucosal surface is continuously exposed to a complex and dynamic community of microorganisms. Although it occupies a relatively small proportion of the intestinal microbiota, *Lactobacilli* has been discovered to have a significant impact on the intestine tract in previous studies. It is undeniable that some *Lactobacillus* strains present probiotic properties through maintaining the micro-ecological balance via different mechanisms, such as mucosal barrier function and barrier immunity, to prevent infection and even to solve some neurology issues by microbiota–gut–brain/liver/lung axis communication. Notably, not only living cells but also *Lactobacillus* derivatives (postbiotics: soluble secreted products and para-probiotics: cell structural components) may exert antipathogenic effects and beneficial functions for the gut mucosal barrier. However, substantial research on specific effects, safety and action mechanisms in vivo should be done. In clinical application of humans and animals, there are still doubts about the precise evaluation of *Lactobacilli*’s safety, therapeutic effect, dosage and other aspects. Therefore, we provide an overview of central issues on the impacts of *Lactobacillus casei* (*L. casei*) and their products on the intestinal mucosal barrier and some diseases and highlight the urgent need for further studies.

## 1. Introduction

The complexity and plasticity of the intestine function have become the focus of current research. The gastrointestinal tract (GIT)can absorb nutrients, transport water and electrolytes, and prevent harmful substances from entering the internal environment from the intestinal cavity [1]. When the intestinal barrier is damaged, it will lead to not only intestinal diseases, but also autoimmune, metabolic and mental health diseases. Therefore, maintaining the integrity of the intestinal barrier is what we must consider. With the development of mucosal immunity research, studies have shown that GIT mucosa can protect the host through innate and acquired immunity [2] and maintain microbiota balance by an immune homeostasis mechanism [3]. From this point of view, the function of GIT mucosa cannot be ignored.

In recent years, significant progress has been made in the symbiotic mechanism between lactic acid bacteria (LAB) and host. The main benefits of probiotics include maintaining intestinal homeostasis [4], improving the intestinal barrier function, immune regulation and the production of neurotransmitters [5]. Probiotics can compete with harmful bacteria for colonization sites in the GIT through a competitive mechanism to expel harmful bacteria. They can also inhibit harmful bacteria by producing lactic acid and changing the pH value in the GIT [6]. Regarding immune regulation, probiotics can produce some biochemical signal substances to improve the body’s immune function. For example, macrophages can recognize and phagocytize harmful bacteria in the colonization sites after enhancing the activity. Some experiments have shown that the pilus-like structure of *L. casei* is easy to bind to mucus-covered mucosal epithelial cells [7] and plays an immunomodulatory role by reducing the expression of pro- inflammatory molecules such as Interleukin-6 (IL-6) and Toll-like receptor 3 (TLR3) [8].

As we know, mucosal infections are one of the biggest health problems worldwide. The mucosal surfaces, thin and permeable barriers, represent an enormous area to be protected. The thin layer of mucosal epithelium can be breached relatively easily. The fragility and permeability of the mucosal surfaces create obvious vulnerability to pathogen infections. The mucosal immune system is an effective defense mechanism, but the barrier function needs to be supplemented by defenses provided by probiotics. For example, *L. casei* fb05 SLP (surface layer protein) could decrease the deleterious effects of *E. coli* ATCC 43893 and *S. enterica serovar* Typhimurium CMCC 50013 on the intestinal tract via reducing pathogen adhesion and inhibiting pathogen-induced apoptosis [9]. *L. casei* and its byproducts alter the virulence factors of foodborne bacterial pathogens [10,11].

There has been deep research on the prevention and treatment by *L. casei* for some diseases, but some overstated propaganda regarding application are questioned. Thus, we review the contribution of *L. casei* and products on the intestinal mucosal barrier and some diseases shown in Figure 1 and highlight the urgent need for further studies.

## 2. *Lactobacillus* Strengthens the Functions of Intestinal Mechanical Barrier

Human and animal mucous membranes are mainly distributed on the inner surface of GIT, the respiratory tract, the urogenital tract and exocrine glands. The mucosa is the first line of defense against pathogens. The GIT is the most exposed part of the body to pathogenic microorganisms, so the development of intestinal mucosal barrier is the most perfect. The intestinal mucosal barrier includes mechanical barrier, mucus barrier, microbial barrier and immune barrier. The normal intestine has a relatively perfect functional isolation zone, which can separate the intestinal cavity from the internal environment and prevent the invasion of pathogenic agents [12]. More recently, the composition of gut and lung microbiota in COVID-19 patients has been altered; that is, dysbiosis. This dysbiosis developed secondary fungal and bacterial infections [13]. Thus, pathogenic microbes and their toxins can disrupt goblet cell function and damage the integrity of the mucus barrier, leading to chronic inflammatory diseases [14]. Probiotic interventions and para-probiotics are useful for gut inflammations by modifying the GIT microbiome, exerting effects on physiology, such as anti-inflammatory responses, and repairing epithelial cell tight junctions [15,16]. The mechanisms of anti-inflammatory action, immune modulation, and modulation of the microbiome have been proved by a lot of research, as mentioned above. However, the role of probiotics in preventing and treating COVID-19 requires further studies [13].

The mechanical barrier is the most important mucosal barrier. The basic structure is complete mucosal epithelial cells, the tight junction between epithelial cells and cell membrane on the surface of epithelium. Intestinal epithelium distinguishes between beneficial and pathogenic microflora through specific molecular patterns of microorganisms, such as pattern recognition receptors (PRRs), which are able to activate specific pathways that lead to inflammation in response to pathogen invasion [17]. Intestinal epithelial cells (IEC) are mainly monolayer columnar epithelial cells composed of absorptive and secretory IEC [18], including enteroendocrine cells, goblet cells and Paneth cells. Mucosal epithelial cells are connected with adjacent cells through various cell connections. The connections from the top of the epithelium to the basement membrane are tight junction (TJ), adhesion junction (AJ), desmosome and gap junction.

### 2.1. Goblet Cells and Paneth Cell Enhanced by L. casei Better Maintain the Host Barrier

Goblet cells are the main contributor of intestinal mucus. Goblet cells enhanced by *L. casei* can better maintain the host barrier between symbiotic microorganisms and the intestine by regulating secretion and removing bacteria, parasites and even fungal infections [19,20]. Some viruses also often employ goblet cells as the target of invasion, such as human adenovirus species C, Enterovirus 71 and human astrovirus VA1 species [21]. Giardia infection can lead to microvilli atrophy, edema and vacuolization of epithelial cells. However, *L. casei* intervention reduced Giardia infection by enhancing goblet cells function [22]. The gut commensal bacteria translocation following antibiotic treatment depended on the colonic goblet cells, which resulted in inflammatory T cell responses to an innocuous antigen [23], hinting that the gut microbiota modulation strategies may prevent this kind of translocation to mitigate antibiotic-associated inflammatory responses. In addition, sentinel goblet cells can quickly respond to pathogens invading the gut through nonspecific endocytosis to protect the mucosa [24].

When feeding ducks with feed containing *L. casei*, the number of goblet cells, villus length and recess depth of duck intestine were all significantly increased [25]. It is impressive that *Lactobacilli* fermented milk also showed the same effect. After taking fermented milk supplemented with *L. casei*, mice were accompanied by hypertrophy and proliferation of Paneth cells and goblet cells [26].

The Paneth cells secreting various antimicrobial peptides and proteins are located at the bottom of the crypts of Lieberkühn in the small intestine. They are professional secretory cells that play an indispensable role in maintaining intestinal homeostasis and regulating microbial flora [27]. Recently, it has been reported that Paneth cells provided the critical developmental and homeostatic signals to the intestinal stem cells [28]. When *L. casei* was used for intervention, a higher percentage of degranulated Paneth cells was observed in the duodenum and jejunum [29], and oral *L. casei* could increase intestinal antibacterial activity [30]. This may be another important mechanism for *L. casei* to maintain intestinal homeostasis and resist virus invasion.

### 2.2. L. casei Increasing Tight Junction Formation and Expression

The function of TJ is to control the free entry and exit of macromolecular water-soluble substances in IEC [31]. TJ is most closely related to the absorption of intestinal nutrients and microbial invasion. TJ is the material basis of selective permeability of epithelial cells, which provides a guarantee for the body by resisting pathogen invasion [32]. TJ proteins are mainly divided into transmembrane proteins and cytoplasmic proteins. Transmembrane proteins include occludin, claudins, and junction adhesion molecules (JAM). Cytoplasmic proteins mainly are comprised of zonula occludens (ZO), cingulin and MUPP1, belonging to the membrane-associated guanylate kinase-like protein (MAGUK) family. Cytoskeleton is another important part of TJ that is a complex structure formed by the interaction of a variety of proteins [33]. TJ regulation involves a variety of signal transduction pathways in vivo, such as the protein kinase pathway, the calcium pathway, the G protein signal regulated protein pathway and the small molecule guanosine triphosphatase (GTP) pathway [34]. The main function of TJ depends on the normal expression and distribution of connexin and cytoskeleton proteins as well as the integrity of the adhesion protein junction. The TJ between intestinal epithelium will be affected by some substances-absorption enhancers, such as EDTA, SDS, glucose, ethanol, etc. When inflammatory intestinal diseases occur, the expression of TJ protein would also be blocked or decreased [35]. If there is sustained hyperosmolarity or a compromise in the protective mechanism, TJ disruption and barrier dysfunction may occur.

Epidermal growth factor (EGF) is an intestinal regulator that plays a vital role in intestinal development and maintaining intestinal barrier. Glutamine (Gln) in an EGFR (EGF receptor)-dependent mechanism significantly up-regulated the levels of occludin, and β-catenin decreased by ethanol feeding [36]. Similarly, probiotics could protect intestinal mucosa by an EGFR-dependent mechanism [37]. Recent findings suggest protective effects of *L. casei* and EGF on epithelium involved in different signaling mechanisms. *L. casei* attenuated osmotic stress-induced TJ disruption and barrier dysfunction by a PKC-dependent mechanism, but independently of EGFR and MAP kinase activities. When EGF and *L. casei* were used together, they were able to prevent the emergency effect of epithelial infiltration through the independent signal mechanism, which was more effective than that of EGF and *L. casei* alone, indicating that *L. casei* might have other protective mechanisms [38].

Mast cells (MCs), widely distributed in the skin and visceral submucosa around blood vessels, can secrete a variety of cytokines and participate in immune regulation [39]. However, uncontrolled MC activation may affect the homeostasis of intestinal environment and further lead to intestinal barrier dysfunction, even GIT disorders. Chunlan Xu et al. reported that when IPEC-J2 cells were infected with enterotoxigenic *Escherichia coli* (*ETEC*) K88, a decrease in cell viability and an increase in permeability occurred. When treated with *L. casei* ATCC393, the mice acquired a significant antagonistic effect on the toxicity of IPEC-J2 caused by *ETEC* K88, and *L. casei* alleviated the down-regulation of the expression levels of TJ proteins occludin and ZO-1 caused by *ETEC* K88. When pretreatment was with *L. casei* ATCC 393, the activation of porcine mucosal mast cells (PMMCs) caused by *ETEC* K88 was reversed, and intestinal barrier dysfunction caused by *ETEC* K88 was also alleviated through the TLRs-signaling-mediated MCs pathway. The number of MCs in the ileal mucosa was reduced. It also significantly inhibited the activation of macrophages and the release of inflammatory factors but improved the expression level of TJ protein [40].

In Yongfei Bai’s experiment, when evaluating the mice intestine transcriptomic data for *L. casei* treatment and gene functions, the expression level of antibacterial peptide REG3a was increased significantly after treatment with *L. casei*. The host stimulated by *L. casei* produced the antibacterial peptide REG3a protein, which significantly promoted the proliferation of cells IPEC-J2 and IEC-6 and the expression of TJ protein ZO-1 and AJ protein E-cadherin [41]. This shows that *L. casei* can protect intestinal mucosal barrier in different ways.

## 3. *Lactobacilli* Inducing the Expression of Mucin to Reduce Adhesion of Pathogens

Mucus is a highly hydrated transparent gel consisting of more than 98% water. Thus, mucus is not easy to study due to the difficulty of observing it under the microscope [42]. Chemical substances secreted by the GIT, such as gastric acid, bile, various digestive enzymes, lysozyme, mucopolysaccharide, glycoprotein and glycolipid, constitute the mucus barrier of the intestinal tract. The complexity of its composition determines the functional diversity.

The key component of mucus is mucin. The function of secretory mucin is to protect and lubricate the GIT [43]. Mucin can make a selective physiological barrier, allowing nutrients to penetrate and macromolecules to pass through. The mucin polymer content is one indicator for evaluating the mucus barrier’s protective function. The mucin-specific functions are to eliminate bacteria, inhibit bacterial adhesion and colonization, clean the intestinal cavity and increase the antibacterial effect of immunoglobulin, and help maintain intestinal homeostasis [42].

When the mucus layer is degraded by the proteases produced by *Escherichia coli*, pathogens can easily adhere to IEC and infect the host, as in the case of *Vibrio cholerae* and the protozoan *Giardia lamblia* in the small intestine [44]. Another way for infection is to invade the host by changing the pH of mucus. For example, *Helicobacter pylori* infects the host when the pH value of mucus rises, leading to the decrease of its viscosity and the increase of the mucus fluidity [14]. Mucopolysaccharides (such as free fucose and sialic acid) as energy and attachment sites promote some pathogen colonization. Pathogens even invade the host by inhibiting mucus production and directly or indirectly regulating goblet cell function and affecting the expression, synthesis and secretion of mucus [14,42]. The role of the mucus barrier also plays an important role in intestinal diseases. During inflammatory bowel disease (IBD), the mucin O-glycosylation profile is changed, the mucin produced by intestinal goblet cells is reduced, and the penetrability of mucus is also changed. The change of the MUC2 structure leads to a loss of mucus viscoelastic properties. The fluidity of bacteria is enhanced, and the number of mucins for degrading bacteria is increased [45].

As early as 1994, J.G. Russeler van Embden and others proved that some *L. casei* strain GG and other strains isolated from a commercial fermented product cannot degrade mucus glycoprotein either in vivo or in vitro [46]. In 2001, A.F. Mattar et al. found that *L. casei* GG could enhance the inhibition of bacterial translocation by upregulating the expression of MUC2 in an in vitro model [7]. Intestinal flora and the mucus layer form a mutually beneficial relationship. Intestinal mucus provides a suitable growth environment for obligate anaerobic bacteria and can promote their growth. As a resident bacterium in human body, *Lactobacillus* induces the production of mucin MUC2, MUC3, mRNA expression and secretion of mucin in IECs (HT-29) to reduce the adhesion of pathogenic *Escherichia coli* [47].

## 4. *L. casei* Changes the Intestinal Microbiota against Some Diseases

With the continuous progress of human society and the change of eating habits, the intestinal microbial community has also changed accordingly. Compared with individuals in the stone age, the intestinal microbiota function of individuals in the modern industrial society has changed significantly, for example, by a significant rise in the number of the genes encoding microbial carbohydrate-active enzymes (cazymes) [48,49]. Intestinal microbiota change with dietary structure, proving that the relationship between intestinal microbiota and host is not just parasitism. In normal wild animals, the number of lymphocytes and the structure of Peyer’s node are very stable. However, in germ-free animals, it shows the opposite situation [50]. The microbiota in the human intestine promote the development of the immune system and protect against the invasion of pathogens, such as *Clostridium difficile* in the colon and cecum [51]. Under normal circumstances, the intestinal microbiota combines with the intestinal mucosa in the form of adhesion or chimerism to make a regular membrane flora, which forms a mutually beneficial intestinal microbiota ecosystem with the host. This microecosystem constitutes the intestine microbial barrier [52]. Normally, intestinal microbiota can secrete antibacterial substances (lactic acid, bacteriocin, etc.), produce short-chain fatty acids (SCFAs), promote intestinal peristalsis, hamper pathogenic bacteria adhesion, and compete for nutrients, which can interfere with and inhibit the vitality and function of pathogenic bacteria through limiting intestinal colonization, dominant reproduction and parenteral translocation.

### 4.1. Changing Intestinal Microbiota

With the development of high-throughput sequencing, the research of intestinal flora on human health has more deeply advanced in recent years, in which the intestinal flora of patients with different diseases can be artificially changed for treatment. Therefore, studying the relationship between intestinal microbiota and diseases is very valuable. *Lactobacillus* can regulate the host health by changing the intestinal microbiota. In a clinical study, *L. casei* LTL1879 isolated from the feces of the long-lived elderly was given to volunteers in powder form. After three weeks of intervention, the number of *Escherichia coli*, *Bacteroides* and *Enterococcus* decreased, but *Clostridium lean*, *Bifidobacterium* and *Lactobacillus* increased significantly; *L. casei* intervention could significantly improve the poor health level of volunteers [53]. This study shows that when taking *L. casei*, there will be no single rise of a certain kind of bacteria but a change in the overall intestinal microbiota. Therefore, the changes are significantly helpful to the host’s health. The same results also appeared in the fecal microbiota transplantation (FMT) experiments of infants with diarrhea. When infants with diarrhea took *L. casei* ZX633 isolated from healthy infants, the abundance of probiotics increased significantly, the abundance of pathogenic bacteria significantly decreased, the level of SCFAs in their bodies was changed, and the symptoms of infant diarrhea were improved [54].

The roles of *L. casei* are not limited to intestinal diseases. It can also regulate intestinal disorders caused by other diseases and improve the clinical symptoms by maintaining intestinal homeostasis. The imbalance of intestinal homeostasis and the impairment of intestinal barrier function have become two of the important pathogeneses of alcoholic liver injury [55], suggesting the gut-liver axis has a key role in the pathogenesis of alcohol dependence. In clinical experiments, the patients with alcoholic liver took *L. casei*, the lipid metabolism was significantly improved, and intestinal microbiota imbalance and alcoholic liver injury were redressed [56]. When *L. casei* CCFM419 was used for treating mice with type II diabetic, all of fasting blood glucose, postprandial blood glucose, glucose tolerance and insulin resistance were improved. Meanwhile, it also reduced the levels of the inflammatory marker tumor necrosis factor-α (TNF-α) and IL-6. The results indicated that *L. casei* CCFM419 adopted the gut flora-SCFA- inflammation/GLP-1 mechanism to ameliorate type 2 diabetes [57]. Due to the existence of the gut-brain axis, probiotics as a therapeutic agent brings new hope for many central nervous system diseases. Recently, studies found that *L. casei* intervention lightened depression in rats induced by unpredictable mild stress (CUMS) and regulated the CUMS by changing intestinal microbiota, which reverses the activities of the ERK1/2 and p38 MAPK signal pathways and various clinical symptoms. The authors believed that this might be related to the composition of intestinal microbiota and mediations of brain-derived neurotrophic factor (BDNF) TrkB signaling [58].

### 4.2. The Role of Intestinal Microbiota in Diseases

The symbiotic effect of intestinal microbiota on hosts not only maintains intestinal homeostasis but also affects some diseases, such as irritable bowel syndrome (IBS) [59], IBD [60], infectious diarrhea [16], cancer, liver disease, diabetes and autoimmune diseases. Bradley et al. discovered that the intestinal microbiota could drive the interference in lung stromal cells α/β signals to improve the ability of these cells to resist influenza virus [61]. The diversity of tumor microbiota can affect the survival of patients with pancreatic ductal adenocarcinoma. Intestinal microbiota colonized in pancreatic tumors modified overall tumoral bacterial composition; transplantation of fecal microflora from long-term survivors into tumor patients changed the tumor microflora, thereby increasing the limitation to tumor growth [62]. The intestinal microbiota also contributes significantly to the development and maturation of the central nervous system, because the communication between the central nervous system and intestine is two-way—the so-called ‘gut-brain axis’. Autoimmune disease (AIDS) is caused by the individual immune system attacking its own tissue. At present, its pathogenesis is still not very clear. However, human microbiota may be the main participant of autoimmunity, because the loss of immune tolerance may occur when the microbial diversity changes [62,63]. With the progress of research into autoimmune diseases and the rapid development of bioinformatics, the relationship between them has gradually surfaced. *L. casei* may become a potential target for treating autoimmune diseases via an intestinal microorganisms change mechanism.

## 5. *L. casei* Enhances Immune Barrier Functions

The mucosal immune system (MIS) is an independent immune system with unique structure-function features and plays critical roles in fighting against infections [64]. The mucosa and lymphoid tissues in the respiratory tract, GIT and urogenital tract are the major sites of the mucosal immune executive function. Intestinal mucosa has a unique tissue structure, such as lymphoid tissue and Peyer’s patch, and special immune cells, such as M cells, intraepithelial lymphocytes (IEL) and IEC. In addition, the number of T cells and B cells in mucosa is much larger than that in immune system. Absorptive epithelial cells can absorb soluble peptide antigens to activate CD8^+^ and CD4^+^ T cells. Professional antigen-presenting cells (APCs) are responsible for macromolecular antigens that cannot be diffused freely in the local tissue. The effect site of intestinal mucosal immunity is divided into IEL and lamina propria lymphocytes. Under the stimulation of antigen, T cells and cytokines, B cell clones become mature IgA-secreted plasma cells. The intestinal microbiota provides both stimulatory and inhibitory signals to the host, ensuring its own survival and contributing to resistance to pathogens [65]. Probiotics have been reported to maintain intestinal balance, to be anti-inflammatory, and to relieve the symptoms of various autoimmune diseases. With the gradual reduction of antibiotic usage, probiotics have been increasingly become considered as a substitute.

### 5.1. History of L. casei as Antigen Delivery Vector

As early as 1986, researchers proposed the effects of an orally-administered mixture of *L. casei* and *L. acidophilus* on the immune system. By continuously feeding Swiss albino mice, it was found that the activities of macrophages and lymphocytes were enhanced, suggesting that *Lactobacilli* might stimulate the host’s immune response [66]. In 1988, Perdigón et al. discovered that the enhancement of this immune response was caused by some metabolites of microorganisms, which proved that living *L. casei* could be used as an immunobiological method to treat diseases [67]. As a vaccine carrier, LAB needs to carry and express foreign antigens. In the 1980s, the genetic modification technology of LAB was becoming mature [68]. In 1990, Gerritse et al. described a transformation system of native LAB in the GIT. When administering the trinitrophenylized (TNP) *Lactobacillus* orally, the anti-TNP serum IgG titers were comparable to or even higher than those of intraperitoneally immunized animals [69]. Thus, the feasibility of using *Lactobacillus* as an antigen delivery carrier was proposed. Rush et al. [70] used the symbiotic *Lactobacillus* from the female genital tract as a carrier to deliver exogenous antigens to the genital mucosal surface, which stimulated a strong local mucosal immune response. In the following years, *L. casei* as an antigen delivery vector became the focus of oral vaccines. In 1995, Erika Isolauri et al. employed *L. casei* combination with DxRRV rhesus human recombinant rotavirus vaccine to improve the titer of the vaccine [71]. In 1999, the experimental results of Maassen et al. showed that recombinant *L. casei* could be used as an oral vaccine to prevent tetanus and induce oral tolerance to intervene in autoimmune disease-multiple sclerosis [72]. So far, more and more *L. casei* delivery systems have been developed and used as microecologics.

### 5.2. L. casei Anchoring Antigens Stimulate the Mucosal Immune Response

The sIgA is the key factor in maintaining symbiotic microflora. The sIgA secreting in mucosa mainly depends on the plasma cells in lamina propria. The specific sIgA in mucosa has been considered as the standard for evaluating mucosal immunity. In addition to inhibiting pathogen adhesion, anti-infection, immune clearance, anti-allergy, promoting natural antibacterial factors and maintaining intestinal homeostasis, sIgA also shapes the composition of the intestinal microbiota [73].

Vaccines are the primary means to prevent viruses, diseases and other diseases. In order to avoid some factors, such as enzyme degradation and less antigen sampling, the antigen delivery system for oral vaccines is constantly being developed in practice. Among them, the mucosal immune delivery system has been widely studied. The mucosal immune delivery system usually refers to the live bacterial or viral carrier carrying the target gene to express the target protein (antigen) in the host in order to transport the antigen into the mucosal sites. This kind of antigen delivery system can directly target the mucosa and express different antigens at the same time to prevent and treat a variety of diseases. The mucosal delivery systems are divided into two categories: one is attenuated pathogens, such as *Escherichia coli*, *Salmonella* and so on. Another is a carrier of probiotics, for instance, LAB. *Lactobacilli* delivery systems have better biosafety and are widely used in practice in China. *Lactobacillus* anchoring antigens can stimulate the local immune response of mucosa and also improve the level of some cytokines such as IL-4, IFN-γ and IL-17 to regulate the differentiation of Th1, Th2 and Th17 cells. *L. casei* has been widely used as an antigen-delivery carrier [74,75].

*Lactobacilli* can be used as a carrier of an active mucosal immune delivery system to transfer antigen. As an antigen delivery carrier, *L. casei* shows many advantages. This will also help to promote the development of oral live vector vaccines. However, the complexity of the mechanism of mucosal immunization brings many challenges to oral vaccines characterized by repeated administration and higher doses of probiotics. It is still challenging to deliver antigens satisfyingly to mucosal induction sites due to the influence of environmental factors of GIT. In addition, ensuring the safety and effectiveness of antigen-delivery vectors is more critical.

The administered dose of the heterologous-expressed *Lactobacillus* vector depends on antigen expression level. It requires multiple administrations to generate a sufficient mucosal immune response. Therefore, selecting an appropriate mucosal immune adjuvant, carrier and antigen delivery system is very important. *Lactobacillus* and its fermentation products enhanced innate immunity and acquired immunity [76]. *L. casei* anchoring foreign antigen can not only improve the host’s innate immunity, but also cause local mucosa and system immune response to achieve a good immune enhancement effect. *L. rhamnosus* is one example of *Lactobacillus* that increased the production of immunoglobulin A (IgA)-secreted B cells in the intestinal mucosa. *L. casei* Shirota activated macrophages to produce interleukin-12 (IL-12) [77]. However, its enhancement effect depends on dose and strains [74]. *L. casei*, as an antigen-delivery carrier, intervenes in some diseases (Table 1).

### 5.3. L. casei Regulates APCs

There is a large population of APCs in the intestines of vertebrates. Macrophages and dendritic cells (DC) belong to antigen-presenting cells. APCs located below the epithelial cell layer can recognize foreign antigens and invading pathogens [81]. M cells and PPs are the primary sites for the uptake and presentation of ingested antigens. M cell is characterized by being conducive to internalizing antigens in the surface of gut epithelial cells and transfer antigen to lymphocytes to induce a local immune response. At present, the primary specific markers of M cells include GP2, PrPc, C5aR, as well as marcks11, m-sec, sgne-1, annexin V, nkm16-2-4, CO1 ligand, ccl9, etc. *L. casei* strain Shirota can be uptaken into M cells in Pyper’s patches, and are then digested to form active components. Moreover, final macrophages or DC obtain the ability to produce a variety of cytokines, resulting in different responses of immune cells [83]. The intervention of *L. casei* protected lupus-prone mice by up-regulating B7-1 and B7-2 of APCs, indicating that these two costimulatory molecules were essential for regulatory T cell (Treg) induction [83].

Probiotics regulating the immune system through APCs have been reported to promote APC maturation. The different probiotics affect APCs in different ways because different probiotic strains stimulate APCs to express different cytokines and surface markers [84]. In Villena and Suzuki’s experiment, *L. jensenii* TL2937 treatment increased the expression of IL-10 in CD172a^+^ cells isolated from PPs [85], suggesting that *L. jensenii* TL2937 could regulate the inflammatory response in the host by regulating APCs.

### 5.4. Regulatory Effect of Probiotics on Macrophages

Macrophages are widely distributed innate immune cells that play an indispensable role in various physiologic and pathologic processes, including organ development, host defense, acute and chronic inflammation, and tissue homeostasis and remodeling [86,87]. There is a large population of macrophages in the intestinal tract of the body, which are derived from monocytes [88,89]. However, intestinal macrophages differ from the other lymphoid system in two respects: the shorter life span, and the reproduction mainly depending on the transformation of monocytes [90,91,92]. Macrophages can prevent pathogen invasion by releasing cytotoxic molecules and proinflammatory cytokines. Bacteria and probiotics can also be sensed by expressing conserved PRRs and microbe-associated molecular patterns (MAMPs), respectively [93,94,95]. Mucosal macrophages can recognize and respond to pathogens, and activated macrophages can migrate to draining lymph nodes [96].

Oral vaccines can be improved by targeting macrophages. RAW264.7 is a macrophage cell line commonly used in the medical research area. *Lactobacillus acidophilus* mtcc-10307 (LA) and *Bacillus Claus* mtcc-8326 (BC) are often used to treat intestinal diseases. According to Pradhan’s experimental results, RAW264.7 cells can be resistant pathogens after being induced by LAB [97]. In T-cell-mediated colitis, after T cells were adoptively transferred in a mouse model, proinflammatory macrophages dominated after 12 h; after three weeks, the proinflammatory macrophages still showed an upward trend [98]. Notably, evidence has also shown that probiotic bacteria possess significant abilities in regulating macrophage polarization both in vivo and in vitro [99]. The polarization of macrophages plays a crucial role in host immune regulation [100]. Macrophages are activated into two extreme subsets: M1 and M2. M1 macrophages can kill intracellular pathogens as potent effector cells, but M2 macrophages can promote wound healing [101]. Some intestinal communities could cause polarization of intestinal M1 macrophages [102]. Butyrate, a metabolite of some intestinal microorganisms, promoted the polarization of M2 macrophages [103]. In the experiment of IBD model mice, it was found that M1, or proinflammatory macrophages, accumulated in the large intestine during colitis. Although the number of M1 macrophages increased, M2 cells still existed and still had the functions of being anti-inflammatory and promoting healing. These data made it clear that intestinal microorganisms played a significant role in promoting the immune function of macrophages (Table 2). As shown in Table 2, many examples of probiotics promote macrophage polarization to achieve immune regulation. Due to the different types of polarization of macrophages, their expressed molecules are also different. Some diseases also lead to macrophage polarization. However, probiotics regulate macrophages not only by promoting their polarization but also by inhibiting their polarization. This mechanism makes the immunomodulatory function of probiotics more extensive and complex.

A recent study found that the immune response in the small intestine induced by *L. casei* could extend to the systemic immune system. At the same time, the activating of macrophages and stimulating of the production of proinflammatory cytokine interferon-γ and TNF occurred [104]. Probiotics can help the host resist *Toxoplasma gondii* infection. In another study, the heat-inactivated *L. casei* IMAU60214 stimulated macrophages to produce NO and cytokines for immune regulation and multifunctional immune activities and enhanced the phagocytosis of macrophages, which mechanism was TLR2-dependent. In this way, *Lactobacillus* promoting the activity of macrophages may be the primary way in early immune response [105]. Macrophages activated by *L. casei* imau60214 induced higher levels of IL-8, IL-6, IL-10 and IL-β, which enhanced the phagocytosis of macrophages, limited the survival of pathogens, and stimulated the expression of the pp65 in the NF-κB signal pathway [106]. In conclusion, *L. casei* can activate the macrophage-associated inflammatory response in vitro through cytokines and signal cascade reaction, improve the innate immune response, and play the function of immune stimulation.

**Table 2 life-12-01910-t002:** Activating the intestinal associated macrophages by *Lactobacillus*.

Strain	Cell Line	Effect	Result	References
*Lactobacillus* brevis G-101	Mouse peritoneal macrophages	M1 to M2	Ameliorates colitis	[107]
*Lactobacillus* plantarum CLP-0611	Mouse peritoneal macrophages	Promote M1 to M2	Ameliorates colitis	[108]
*Lactobacillus* acidophilus LA1	Mouse peritoneal macrophages	Induce M2	Suppresses intestinal inflammation	[109]
*L. casei* cell wall extract	RAW 264.7 cells	Induce M2	Enhanced surface expression of dectin-1 and TLR2	[110]
*L. casei* HY7213	Mouse peritoneal macrophages	Significantly restored phagocytosis activity	Enhanced surface expression of dectin-1 and TLR2	[111]
*L. casei* 1–5	Mouse peritoneal macrophages	Macrophage activation	May be involved in the prevention of pathogenic *E. coli* infection	[112]

### 5.5. Regulation of DC and T Cells by L. casei

DC is the most potent professional antigen-presenting cell in the body. They are widely distributed in mucosal epithelium and subepithelial connective tissue. DCs uptake pathogenic microbial antigens and present antigens to lymphocytes to activate an immune response. The main specific markers of DC cells are CD11c and MHCII molecules. Macrophages also have such surface markers, but the expression is very low, and that in DC cells is very high. Mucosal DCs at different locations have different phenotypes and functions.

DC that induce mucosal immune and controlled pathogens can be divided into conventional DC (CDC) and plasmacytoid DC (PDC). DCs also can maintain tolerance and control allergies. The trafficking, specialization, plasticity and crosstalk of DC play an important role in shaping the mucosal immune response. The pathogenic mechanism of some viruses decreases T cell immune response by inhibiting the differentiation of DC. Some viruses can also achieve the purpose of autoimmune escape by inhibiting the maturation of DC. DC-homing could be promoted via inducing the expression of CCR7 on DC [113]. A bromodeoxyuridine pulse-tracking experiment showed that most of the CD103^+^ DCs in mesenteric lymph nodes (MLN) migrated from the lamina propria of small intestinal mucosa. This intestinal DC subset preferentially generates Treg in mice. However, most CD103^−^ DC in MLN is an inherent cell population in lymph node homeostasis [113]. In patients with UC, intestinal DC subsets were skewed in the wake of a loss of CD103^+^ lymph-node homing DC; however, when treated with *L. casei* Shirota, UC patients partially recovered their ability for imprinting homing molecules on T cells and generating interleukin-22 by stimulated T cells [114]. The stimulation ability of DC to T cells in UC patients decreased. However, when treated with *L. casei* Shirota, the stimulation ability in UC mice was significantly improved [115].

The migration of DC is the premise of their function. DC first recognizes pathogens through mucosal PRRs, then starts a series of maturation and transmission for cell mobilization and interstitial migration, entry into the afferent lymphatics, and transit via the lymph to secondary lymph nodes [116]. The antigen is then presented to naive lymphocytes to induce a series of immune responses, such as secreting cytokines and promoting the differentiation of IgA-switched B cells. In addition to the migration of DC, it is noteworthy that mucosal DC imprints specific homing patterns on lymphocytes. Therefore, increasing the vaccine titer by improving cell migration is one of the focuses of current research. DC migrates after ingestion of antigen or the action of some stimulating factors, which is strictly controlled by the expression of CCR7.

Recently, with the rise of oral mucosal immune vaccine of LAB, significant progress has been made using LAB as a carrier to deliver vaccine antigen to the intestinal mucosal system for inducing a protective immune response [117,118]. Moreover, an oral vaccine targeting intestinal mucosal DC improved the delivery efficiency of vaccine antigen to MIS [119]. In Xiaona Wang et al.’s experiment, *L. casei* were used to deliver the DC-targeting peptide (DCpep) fused with the PEDV core neutralizing epitope (COE), which titers were significantly increased [120]. Using *L. casei* as an antigen delivery vector, targeting DC antigens can effectively activate DC in Peyer’s patches (Table 3).

Intestinal mucosa contain a large number of T cells, mainly distributed in gut-associated lymphoid tissue (GALT), lamina propria and epithelial cells. Naive T cells in gut epithelial cells receive antigens from APCs, then migrate to MLN, differentiate into effector T cells or memory T cells in MLN, enter the system circulation through the output lymphatic vessels, and finally return to the lamina propria in the gut. T cells in intestinal mucosa play an important role in maintaining intestinal homeostasis, and their unique phenotype and diversity reflect their powerful function in mucosal immunity [123]. To date, many studies on *L. casei* show a benefit in disease recovery through T cells, which is summarized in Table 4.

## 6. The *Lactobacillus* Treatment of Some Immune Diseases

### 6.1. Food Allergy

In recent years, with the improvement of living standards, dietary levels have also gradually increased, and food allergies have gradually become threats. The incidence rate of food allergy caused by shellfish is also increasing year by year, along with the increasing love for seafood. Tropomyosin (TM) is the main culprit of allergic reactions caused by shellfish, one of the three major food allergens (peanuts, milk and shellfish) [132]. In recent studies, BALB/c mice fed by TM were treated with *L. casei* Zhang, which changed the development and function of DC, T cells and B cells via activating the NF-κB signaling pathway; finally, they led to the transformation of the TM-specific antibody subtype into tolerance mode [133]. In a clinical trial, *Lactobacillus rhamnosus* benefited children with atopic dermatitis and cow’s milk protein allergy [134]. *L. casei* 1134 fermentation products can significantly reduce the antigenicity and allergenicity of *α*-lactalbumin, *β*-lactoglobulin, *α*-casein and β-casein in milk [135], suggesting this may be one of the ways for *L. casei* to intervene in allergies.

### 6.2. Systemic Lupus Erythematosus

Systemic lupus erythematosus (SLE) is a typical systemic autoimmune disease characterized by highly active immune cells and abnormal antibody response. In the past few years, the relationship between SLE and intestinal microorganisms has been continuously explored. Some studies have shown that the clinical symptom of SLE patients returns to normal after being treated by a variety of microorganisms. Part of the reason may be that the host’s immune response and immune protection mechanism change when microorganisms infect the host [136]. In a classic mouse model of SLE (MRL/lpr mice), the abundance of LAB in the intestine decreased significantly compared with the healthy group [137]. When intestinal microorganisms in SLE patients treated with prednisone are transplanted into MRL/lpr mice, they can reduce the clinical symptoms of MRL without the side effects of prednisone. It is proved that intestinal microorganisms can be used as a potential way to treat SLE [138], as found in subsequent studies. In other SLE models (NZB/W F1 mice), the intestinal microflora showed different diversities at different stages of the disease [139]. Some studies have elucidated that feeding *L. casei* B255 before establishing BWF1 mouse model can delay the onset of lupus. The mechanism study found that *L. casei* significantly up-regulated the production of IL-10, which plays a significant role in the induction of Treg cells, which protective mechanism was related to the up-regulation of B7-1 and B7-2 by antigen-presenting cells [140].

### 6.3. Rheumatoid Arthritis

Rheumatoid arthritis (RA) is a chronic immune-mediated inflammatory disease characterized by joint swelling, injury and synovitis. Intestinal microbial imbalance is one of the causes of RA. As a probiotic, *L. casei* has outstanding performance in regulating intestinal homeostasis and anti-inflammatory cytokines, indicating it is possible to use *L. casei* as a potential treatment. *L. casei* can promote the production of SCFAs that can control different innate immune activities and regulate external inflammation and oxidative stress. For example, butyric acid can inhibit the proliferation of antigen-specific B cells and plasmablastic cells and the production of innate killer T cell (NKT) cytokines. This cell subset is mainly related to joint inflammation and tissue degeneration [141]. Collagen-induced arthritis (CIA) may be an autoimmune disease dominated by T lymphocytes. Collagen protein is an agent commonly used to set up RA models in the laboratory. Studies have shown that oral administration of *L. casei* reduced the production of abnormal antibodies from type II collagen immunity, which was a sign of up-regulation of FoxP3-positive Treg cells. The anti-inflammatory and antibacterial properties of probiotics can be achieved by regulating local and systemic metabolites by SCFAs, histamine and adenosine [142]. Probiotics inhibit cyclooxygenase-2 (COX-2) and reduce chemokine to exert an anti-inflammatory effect. Another study found that the CIA cytokine profile also changed after *L. casei* treatment and improved the phenomena of joint edema, cartilage degradation and lymphocyte infiltration in clinical symptoms [143]. SCFAs can also regulate leukocyte function, including the production of inflammatory cytokines and chemokines, such as IL-10 and IL-2 [144]. The signal cascade generated by the attachment of *L. casei* to epithelial cells may regulate the immune response to reduce oxidative stress and the risk associated with RA [145]. *L. casei* also plays a role in improving CIA by intestinal flora. The studies found that increasing *Lactobacillus* in the intestine inhibited inflammatory cells through releasing antibodies and antibiotic stimulants, changed Th1/Th17 response, improved CIA symptoms, and enhanced the activity of phagocytosis and lymphocytes [146,147,148]. In clinical experiments, *L. casei* could reduce cytokines IL-17, IL-1, IL-6 and tumor necrosis factor-α to inhibit the occurrence of arthritis [149]. Moreover, *Lactobacillus* elevated the levels of natriuretic and reactive oxygenated species (ROS) and reduced RA-related problems [150,151].

## 7. Summarizing the Studies of the Gut Mucosal Barrier and Diseases in Animal, Human and In Vitro

Some current research on *Lactobacilli* functions was designed in an in vitro model. These results are mainly focused on the increased expression of TJ proteins and the activation of their related pathways to maintain the function of the intestinal mucosal barrier and further resist the invasion of pathogens (Table 5). However, the actual intestinal environment is complex and changeable; therefore, it is necessary to use the in vivo experiment to detect the role of *L. casei*.

*Lactobacillus* has generated a rich diversity group in probiotic organisms. Each *Lactobacilli* strain has unique genetic traits, so the features from one bacterial strain or species cannot necessarily be applied to a close relative. More specific functions vary in different *L. casei* strains, which have proved different biological effects in animal models. The relationship studies among *L. casei*, intestinal mucosal barrier and disease are shown in Table 6.

In human experiments, *L. casei* showed different effects, not only improving the level of intestinal disorders in patients with alcoholic liver disease, but also improving lipid metabolism. Like the results in animal experiments, *L. casei* can regulate the secretion of proinflammatory factors and improve intestinal inflammation. It can also treat allergic diseases through immune tolerance. Although there is no clinical trial on the effects of different *L. casei* strains, some clinical results have demonstrated that different strains exhibit different effects, suggesting that different biological properties can lead to different clinical effects (Table 7).

## 8. Probiotic Pretreatment for Entering the Intestine

The relationship between intestinal barrier and microbiota is complex. With the development of the broad application of probiotics, their safety has been paid great attention. There are also many risks in the process of *Lactobacillus* use, such as infection caused by bacterial translocation, which is common in children and patients with a decreased body defense function and unbalanced neonatal intestinal microbiota [165]. Therefore, it is necessary to ensure the probiotic properties of pre-treatment *Lactobacilli* in processing, handling and storage in shelf-life. Probiotics are defined as “live microorganisms, which, when consumed in adequate amounts, confer a health effect on the host” [166,167]. However, recent evidence indicates that the cell fragments of dead bacteria still retain biological activity and can induce an immune response similar to living bacteria [168,169]. Recently the term “para-probiotics” was referred to as nonviable or inactivated bacterial cells and crude cell extracts [169,170,171].

### 8.1. Advantages of Heat-Killed Bacteria

Inactivation is to kill microorganisms by physical or chemical methods. In the pretreatment of probiotics, we usually use high temperatures to make probiotics no longer have the ability to grow and reproduce but maintain the basic morphology of cells. Although probiotics have lost the ability to reproduce, they still have immunogenicity and can still activate the human immune system. A large number of experiments have proved that its role in immunogenicity, promoting nutrient absorption and maintaining intestinal homeostasis will not be reduced or lost with the loss of activity [172]. For example, SCFA supplements in feed boost the immune response [173,174] and decrease pro-inflammatory cytokine expression in challenged broiler models [175]. Probiotic inactivation is often used in the development of probiotic preparations. According to the research of Piqué et al. 2019, the inactivated probiotics and their metabolites still have the function of directly killing pathogenic microorganisms and can adhere to the IECs of the host, forming biofilms to protect the intestine [176,177].

Since 2004, there has been increasing evidence suggesting that some health benefits of active probiotics are not strictly associated with their viability. According to Siciliano et al., the para-probiotics molecules present on the cell surface, such as peptidoglycan, teichoic acid, cell wall polysaccharides, cell surface-associated proteins, etc., mediated the similar effects to viable probiotics [171]. Very recently, the heat processing methods for *L. casei* and *L. paracasei* include Ohmic heating (8 V/cm, 95 °C for 7 min, 60 Hz) [178], and heat-killed in water (65 °C, 60 min) [179] and 85 °C, 20 s [180].

The heat-inactivated probiotics will not be changed due to environmental factors, but the living bacteria will have changed because of the influence of temperature and concentration. The heat-inactivated probiotics do not need refrigeration or other special protection in the transportation process, which dramatically reduces the use cost. More particularly, the heat-inactivated probiotics can be used in combination with antibiotics to produce synergistic effects. In some treatment schemes, heat-inactivated bacteria are often taken during the administration of antibiotics, which can promote the homeostasis of intestinal microorganisms. Heat-inactivated probiotics will not produce intestinal microbial imbalance and lead to the outbreak of phage pollution [176] because they have lost their reproductive ability. This ensures the safety of the host, and even patients with immunodeficiency will not have side effects. Amazingly, heat-inactivated probiotics are far from losing their ability for competitive rejection, and the ability for adsorbing some toxins becomes stronger. For that reason, the heat-inactivated probiotics can be used in low-immunity adults and children [171,178]. Malnutrition is usually related to immune disorders. Clinically, malnourished children often have lower functions in cellular and humoral immune, so bacterial and viral infections are common complications in this condition. A recent study on malnourished children reported that heat-inactivated *L. casei* IMAU60214 increased phagocytosis, bactericidal capacity and regulation of the immune function of macrophages [46].

More and more studies have shown that oral heat-inactivated probiotics, such as *Enterococcus faecalis* EC-12, *L. gasseri* CP2305, and *L. kunkeei* YB38, affect the composition of intestinal flora [181,182,183]. The experimental results showed that the immune effects of some strains treated by heat were higher than those of live strains. For example, in a multispecies combination of LAB (MLAB) composed of four LAB strains, compared with living MLAB, the expression level of IL-12 induced by heat inactivation bacteria was higher [184].

### 8.2. Effects of Living L. casei on Host

LAB can be colonized in various animals and plants and coevolve with the host. A large number of experiments have proved that LAB can further regulate the metabolic process of the host and other microorganisms by colonizing in the mucosal site of tissue of the GIT [69] and affecting the microflora composition in the host. At present, LABs are used to make various probiotic products for treating various diseases, not just GIT diseases. For instance, the effects of *L. casei* preparations are widespread. The *L. casei* group had the potential to reduce the risk of allergy [185]. *L. rhamnosus* JB1 reduced anxiety and depression in mice by affecting the expression of GABA receptors in the brain [186]. The gut microbiota produces some important functional bioactive molecules, for instance, signaling molecules for the neuronal circuits, the activator of immune system and its response to a possible pathogenic threat, gut hormones and short-chain fatty acids. These products exert a pivotal role in neurogenesis, mental and cognitive development, emotions, and behaviors, and in the progression of neuropsychiatric illnesses [187]. *L. casei* preparations were also combined with other probiotics to achieve satisfactory results. The rehospitalization rate of patients with mania was decreased when taking tablets of *L. casei* group and *Bifidobacterium* Bb12 as adjuvant treatment [188]. *L. casei* CRL431 positively affected some biomarkers related to obesity [189] and might affect obesity in this way. Cancer has always been a global problem. Probiotics are now being studied as adjuvant therapy or microbial therapy [190]. It was found that *L. casei* ATCC393 had effects against the proliferation of colon cancer cells. The tumor volume was reduced by about 80% in vivo [191]. The cell-free supernatants of *L. casei* and *L. rhamnosus* could diminish the invasive ability of metastatic tumor cell lines in vitro [192]. In brief, *L. casei* provides a new idea for anti-cancer.

## 9. Summary and Outlook

*L. casei* and its derivatives have unique advantages in maintaining the intestinal barrier. However, the colonization time in the intestine is still one of the problems we face now. Our previous studies have shown that feeding heterogeneous *L. casei* to mice for five days can keep the abundance of flora increasing for about a week and then revert back to the pre-intervention composition [172]. Therefore, indigenous *L. casei* maintaining a high flora abundance in the intestine and *Lactobacilli* dietary supplements may be a practical solution approach in future practice. Among the future directions, it is necessary to consider the intervention duration when studying the effects of dietary strategies, treatments or interventions on the gut microbiota.

More and more evidence has demonstrated that feeding *L. casei* significantly improved the composition of intestinal microorganisms, inhibited pathogenic bacteria and improved the abundance of probiotics. However, the interaction between *L. casei* and other probiotics is not very clear. For example, some probiotic combinations can prevent *Clostridium difficile* infection, but their interactions have not been completely discovered, and whether they will have some long-term effects is unknown [193].

In recent years, the emergence of drug-resistant bacterial pathogens has been a major public health security challenge. According to our previous research, *L. casei* could stimulate the production of antibacterial peptides in the host intestine. Antibacterial molecules in the host intestine, including antibacterial peptides and bacteriocins produced by intestinal epithelial cells, as well as non-bacteriocin molecules produced by gram-positive and Gram-negative bacteria, may be currently the first choice as antibacterial drugs. We also found that *L. casei* stimulated the intestine to produce antibacterial peptides, in order to resist the invasion of *ETEC* and maintain the integrity of the intestinal mucosal barrier [41]. Although there has deep research on the prevention and treatment of some diseases by *L. casei* and derivatives, we still need to clarify whether there are some unknown hidden dangers apart from constipation. Therefore, it is highly necessary to gain a deep understanding of the action mechanism of *L. casei* as well as to get more preclinical and clinical study evidence. These further studies will also help to develop new treatment schemes.

It is unquestionable that the application of *Lactobacillus* and its derivatives has broad prospects, especially for drug-resistant therapeutics, immunoregulatory functions, limiting the infectivity of GIT pathogens, restoring intestinal microecological balance and improving neurological diseases. Nevertheless, more precision in dosage, efficacy and safety data as therapeutic drugs are also needed. The traditional drug development paradigms in preclinical and clinical studies are still lacking. In the future, some factors should be considered to better address issues related to prebiotics and para-probiotics as treatment agents, including efficacy (especially antiviral effect), safety, dosage and quality assurance, etc. The side effects of long-term use as a food additive still need more research and clinical evidence. Regarding mucosal immunity, future studies of *Lactobacillus* and its derivatives should focus on the precise specific effect in humans and animals, elucidate the mechanisms of action in vivo, and link this to more advanced evaluation methodologies.

## Figures and Tables

**Figure 1 life-12-01910-f001:**
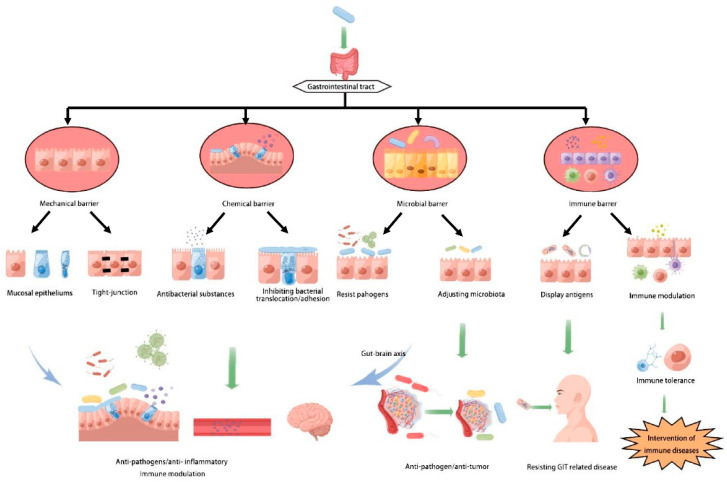
Roles of *L. casei* in intestinal mucosal barrier.

**Table 1 life-12-01910-t001:** The immunomodulatory function of *L. casei*.

Strain	Expressed Protein	Experimental Object	Experimental Phenomenon	Symptoms Improvement	References
*L. casei* ATCC 393	OMP19 Brucella protein	Mice	Increase IgA, sIgA, IFN-γ, IL-2, IL-4	Protection against brucellosis	[78]
*L. casei* 393	COE antigen of PEDV M cell-targeting peptide Co1	Mice	Increase IgG, sIgA, IFN, IL-2, IL-4, and proliferation of Th2-type cells.	Promising oral vaccine candidate for PEDV	[79]
*L. casei* CC16	AcrV secreted protein of the *A. veronii* TH0426 strain	Crucian carp	Increase IgM, IL-10, IL-1β, IFN-γ, TNF-α	Promising candidate for an oral vaccine against *A. veronii*	[80]
*L. casei* ATCC 393	AHA1-CK6 and VP2 of Infectious pancreatic necrosis virus	Rainbow trout	Increase IgM and skin mucus IgT, IL-1β, IL-8 and TNF-α	Induce mucosal immune response and prevent IPNV infection	[81]
*L. casei* CICC 6105	*ETEC* 987P fimbrial protein	Mice	Increase IgG, sIgA, enhance T-cell proliferation	Induce mucosal immune response and prevent *ETEC* infection	[82]

**Table 3 life-12-01910-t003:** The protection function of DC-targeting recombinant *L. casei*.

Strain	Expressed Protein	Experimental Object	Experimental Phenomenon	Symptoms Improvement	References
*L. casei* strain ATCC 39392	DC-targeting peptide fused with PEDV COE antigen	Piglet	IgA ↑, IgG ↑Th1 ↑	The efficacy of protecting piglets from PEDV infection was 60%	[121]
*L. casei* strain W56	BVDV glycoprotein E2 with DC-targeting peptide	BALB/c mice	Activate DC in PPs, lymphoproliferative responses ↑, Th1-associated IFN-γ ↑, Th2-associated IL-4 ↑, sIgA ↑, IgG ↑	Induce anti-BVDV mucosal, humoral, and cellular immune responses	[122]
*L. casei* ATCC 393	COE-Col-DCpep fusion genes	BALB/c mice	sIgA ↑, Th1-related IFN-γ ↑, Th2-related IL-4 ↑	Efficiently induce anti-PEDV mucosal, humoral, and cellular immune responses	[119]

Note: the ↑ represents an increase.

**Table 4 life-12-01910-t004:** *L. casei* interferes with some diseases through T cells.

Strain	Experimental Object	Experimental Phenomenon	Symptoms Improvement	References
*L. casei* BL23	Mice	Induces CD4 + FoxP3 + Treg cells	Inhibit intestinal inflammation	[124]
*L. casei* BL23	Mice	Modulation of regulatory T-cells toward a Th17-biased immune response	Reduces 1,2-dimethylhydrazine -associated colorectal cancer	[125]
*L. casei* CCFM107	Rats	Balancing Treg/Th17 and Modulating the Metabolites and Gut Microbiota	Alleviates collagen-induced arthritis	[126]
*L. casei* LH23	Mouse	Increase in CD3 + CD4 + CD25 + Treg reducing numbers of macrophages (CD11b + F4/80+) and their secreted inflammatory cytokines	Alleviates collagen-induced arthritis	[127]
Exopolysaccharide produced by *L. casei*	Mouse	Increase in CD3 + CD4 + CD25 + Treg reducing numbers of macrophages (CD11b + F4/80+) and their secreted inflammatory cytokines	Improves intestinal mucosa immunity.	[128]
*L. casei* 393	Murine	Modulated splenic CD4+, CD8+, and NK T cell subpopulations. Cytokine homeostasis and the maintenance of a healthy T cell subpopulation dynamic	Promotes immune response	[129]
*L. casei* Lbs2	BALB/c mice	Polarized Th0 cells to Treg cells, increased the frequency of FoxP3+Treg cells	Intervention colitis	[130]
*L. casei* ATCC 393	Mouse	Induced potent Th1 immune responses and cytotoxic T cell infiltration in the tumor tissue	Alleviate colon carcinoma	[131]

**Table 5 life-12-01910-t005:** Studies of *L. casei* on intestinal mucosal barrier and diseases in vitro.

Strain	Experimental Object	Model	Experimental Phenomenon	Symptoms Improvement	References
*L. casei* ATCC 393	PMMCs	*ETEC*-K88 infection	Induced the activation of PMMCs	Alleviated the intestinal mucosal injury caused by *ETEC* K88	[152]
*L. casei* ATCC 393	Mast cells	*ETEC*-K88 infection	*L. casei* ATCC 393 via TLRs signaling pathway prevented intestinal mast cell activation by *ETEC* K88	Alleviated intestinal barrier dysfunction caused by *ETEC* K88	[40]
*L. casei* LC01	Human IEC lines	Human IEC lines (HCoEpiC, C1388)	miR-144 ↓, FD4 ↓, OCLN ↑, ZO-1 ↑	Regulates intestinal permeability of IECs	[153]
*L. casei* DN-114 001	Caco-2	TNF-α or IFN-γ-induced epithelial barrier dysfunctions	Trans-epithelial resistance ↑, ZO-1 ↑, TLR2 ↑, p-Akt ↑	Prevents cytokine-induced epithelial barrier dysfunctions in IECs	[154]

Note: the ↑ represents an increase, the ↓ represents a decrease.

**Table 6 life-12-01910-t006:** Research of *L. casei* on intestinal mucosal barrier and diseases in animal.

Strain	Experimental Object	Model	Experimental Phenomenon	Symptoms Improvement	References
*L. casei DBN023*	White Leghorn chicks	*S*. *pullorum* infection.	IL-22 ↑, activate the Wingless-Int pathway, ZO-1 ↑, Claudin-1 ↑	Regulate the intestinal inflammatory response of chicks infected with *S. pullorum*	[155]
*Lactobacillus* G15 *Lactobacillus* G14	Male Wistar rats	Type 2 diabetes rats	Hb1Ac ↓, IL-1β ↓, IL-8 ↓	Alleviated inflammatory status and islet β-cells dysfunction	[156]
*L. casei* DN-114 001	BALB/c mice	IBD	Significant protection against increased intestinal permeability, ZO-1 ↑, TNF-α ↓, IFN-γ ↓, IL-10 ↓	Prevent the development of severe forms of intestinal inflammation	[157]
*L. casei* CRL 431	BALB/c mice	*Salmonella enteritidis serovar Typhimurium* infection	Activated the macrophage phagocytic activity, decreased the neutrophil infiltration, increased the number of IgA + cells in the lamina propria of the small intestine	Decreased the severity of the infection with *Salmonella enteritidis serovar Typhimurium*	[158]
*L. casei*	Pregnant Sprague Dawley female rats	Postpartum depression rat	Reversed the changes of BDNF, N-methyl-D-aspartic acid receptor 1 (NR1), ERK1/2, and monoamines in the brain of PPD rats	Improved depressive-like behaviors, intestinal microflora, and oxidative stress in PPD model rats	[159]
*L. casei* CRL 431	BALB/c mice	Endotoxemia model	TNF-α ↓, IL-6 ↓, lower activation of the coagulation system, fast systemic restoration of factors VII and V coagulation factors and antithrombin levels	*L. casei* CRL 431 to regulate the immuno-coagulative response	[160]

Note: the ↑ represents an increase, the ↓ represents a decrease.

**Table 7 life-12-01910-t007:** Research of *L. casei* on intestinal mucosal barrier and diseases in human.

Strain	Experimental Object	Experimental Phenomenon	Symptoms Improvement	References
*L. casei Shirota strain*	Patients with alcoholic liver	Significant increase in the amount of *Lactobacillus* and *Bifidobacterium*	Improve lipid metabolism and regulate intestinal flora disorders	[56]
*L. casei*	Patients with pulmonary tuberculosis	Levels of TNF-α, IL-6, IL-10, IL-12 decreased, Levels of maresin 1, phosphatidylserine, pyridoxamine, phosphatidylcholine, L-saccharopine increased	Significantly modulate inflammatory cytokines and metabolites	[161]
*L. casei* variety rhamnosus	Children with acute diarrhea	Fecal sIgA levels were up-regulated, and concentrations of fecal lactoferrin and calprotectin were significantly downregulated	*L. casei* variety rhamnosus may be a useful supplement for application in children during acute diarrhea	[162]
*L. casei*	Patients with Type 2 diabetes mellitus	Affected SIRT1 and fetuin-A levels in a way that improved glycemic response	Affecting the SIRT1 and fetuin-A levels introduces a new known mechanism of probiotic action in diabetes management	[163]
Six viable probiotics of 3.0 × 10^10^ cfu *Lactobacillus* and *Bifidobacteria* strains	Patients with colorectal cancer	Significant reduction in level of pro-inflammatory cytokine, TNF-α, IL-6, IL-10, IL-12, IL-17A, IL-17C and IL-22	Probiotics may modify intestinal microenvironment, resulting in a decline in pro-inflammatory cytokines	[164]

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
