# Peer review of "Contribution of Lactobacilli on Intestinal Mucosal Barrier and Diseases: Perspectives and Challenges of Lactobacillus casei"

_life, 2022, doi:10.3390/life12111910_

Round 1

Reviewer 1 Report

Qin et al, well summarized the effects of L. casei strains from varioius points of view in tables. However, the organization of the main text is poor, which makes this review article difficult in reading and understanding. It is required to re-consider of the sub-titles, and text organization. Also, the authors may provide figures for readers to help understanding.

Author Response

Dear reviewer

Reviewer 2 Report

The manuscript by Da Qin et al. summarized the contribution of Lactobacillus casei to intestinal mucosal barrier and diseases. This is an old topic and the manuscript is poorly written. The authors must critically revise the manuscript before resubmit to the journal. 

1, The English of the manuscript must be critically refined. So many eroors and typos! For example, Goble cell ect. 

2, At least one or two figures should be added to illustrate the main findings. The authors must provide new ideas and new findings even in a review paper. 

3, In the Summary and outlook section, limitations of the current studies must be critically discussed and directions for the future development must be provided and further discussed. 

4, Another two or three tables summarizing the in vitro, animal and human studies on the contributions of Lactobacillus casei to intestinal mucosal barrier and diseases must be added and discussed. 

Author Response

Dear reviewer.

Reviewer 3 Report

In general, this review article is really interesting and well written. My comments aim to increase the scientific soundness and clarity of it.

Some language ambiguities and errors can be found, so I highly advise a help of native-speaker.

Line 2 – The topic of this article is not new and role of probiotics has been presented several times before. There are many more comprehensive articles dealing with the same or very similar subject. See for instance:

a)      Maldonado Galdeano C, Novotny Nuñez I, Carmuega E, de Moreno de LeBlanc A, Perdigón G. Role of probiotics and functional foods in health: gut immune stimulation by two probiotic strains and a potential probiotic yoghurt. Endocr Metab Immune Disord Drug Targets. 2015;15(1):37-45.

b)      Camilleri M. Human Intestinal Barrier: Effects of Stressors, Diet, Prebiotics, and Probiotics. Clin Transl Gastroenterol. 2021 Jan 25;12(1):e00308.

c)      Bron PA, Kleerebezem M, Brummer RJ, Cani PD, Mercenier A, MacDonald TT, Garcia-Ródenas CL, Wells JM. Can probiotics modulate human disease by impacting intestinal barrier function? Br J Nutr. 2017 Jan;117(1):93-107.

d)      Plaza-Díaz J, Solís-Urra P, Rodríguez-Rodríguez F, Olivares-Arancibia J, Navarro-Oliveros M, Abadía-Molina F, Álvarez-Mercado AI. The Gut Barrier, Intestinal Microbiota, and Liver Disease: Molecular Mechanisms and Strategies to Manage. Int J Mol Sci. 2020 Nov 7;21(21):8351.

To make this article more interesting to a reader I suggest the authors to precisely and clearly define the target question of this review. In other case this article is without novelty.

Line 13 – Simple summary and abstract are nearly the same. I think simple summary should present the idea of the work without technical details but not to duplicate the abstract.

Line 60 – please explain what IL-6 and TLR3 stand for. Also some other acronyms in the text need explanation.

Line 116 - From histological point of view there are only four kinds of tissues: epithelial, connective, muscular and nervous. Therefore such terms as “mucosal tissue” (line 116) or “lymphoid tissue” (line 372, 616) are not justified.

Line 134, 135 – digestive tract and GIT are not the same !!! In this case please change to GIT.

Line 371 – please use GIT as previously abbreviated

Author Response

Dear reviewer

Reviewer 4 Report

Dear authors,

I revised the manuscript with ID life-1990246 dedicated to the study of  Lactobacillus casei in relation to the intestinal mucosal barrier and associated diseases. The topic of the manuscript falls within the aim of the journal, but the focus of the manuscript is not clearly pointed out. Thus, some aspects must be clarified before the manuscript is considered for possible publication. Additional comments can be seen in the attachement.

Regards,

Author Response

Dear reviewer

Round 2

Reviewer 1 Report

The manuscript is improved significantly.

Author Response

We do very much appreciate your comments. 

Reviewer 2 Report

The authors have revised the manuscript accordingly. However, the English of the manuscript must be further improved. 

Author Response

Dear reviewer

Reviewer 4 Report

Thank you for upgrading the manuscript.

Author Response

(The authors gave the same response as above.)
